

# Clustering has a meaning: optimization of angular similarity to detect 3D geometric anomalies in geological terrains

Michał P. Michalak[1,2], Lesław Teper[1], Florian Wellmann[3], Jerzy Żaba[1], Krzysztof Gaidzik[1], Marcin Kostur[4], Yuriy P. Maystrenko[5], Paulina Leonowicz[6]

[1]Institute of Earth Sciences, Faculty of Natural Sciences, University of Silesia in Katowice, Będzińska 60, 41-205 Sosnowiec, Poland, ORCID: https://orcid.org/0000-0002-1376-235X
[2]Faculty of Geology, Geophysics and Environmental Protection, AGH University of Science and Technology, Mickiewicza 30, 30-059 Cracow, Poland
[3]Computational Geoscience and Reservoir Engineering, RWTH Aachen, Wüllnerstr. 2, 52056 Aachen, Germany
[4]Faculty of Science and Technology, University of Silesia in Katowice, 75. Pułku Piechoty, 41-500 Chorzów, Poland
[5]The Geological Survey of Norway (NGU), Leiv Eirikssons vei 39, 7040 Trondheim, Norway
[6]Faculty of Geology, University of Warsaw, Żwirki i Wigury 93, PL-02-089 Warszawa, Poland

*Correspondence to*: Michał P. Michalak (michalmmichalak@us.edu.pl, michalm@agh.edu.pl )

**Abstract.** The geological potential of sparse subsurface data is not being fully exploited since the available workflows are not specifically designed to detect and interpret 3D geometric anomalies hidden in the data. We develop a new unsupervised machine learning framework to cluster and analyze the spatial distribution of orientations sampled throughout a geological interface. Our method employs Delaunay triangulation and clustering with the squared Euclidean distance to cluster local unit orientations/attitude which results in minimizing the within-cluster cosine distance. We performed the clustering on two representations of the triangles: normal and dip vectors. The classes resulting from clustering were attached to a geometric centre of a triangle (irregular version). We developed also a regular version of spatial clustering which allows to answer whether points from a grid structure can be affected by anomalies. To illustrate the usefulness of the combination between cosine distance as dissimilarity metric and two cartographic versions, we analyzed subsurface data documenting two horizons: 1) the bottom Jurassic surface from the Central European Basin System (CEBS) and 2) an interface between Middle-Jurassic units within the Kraków-Silesian Homocline (KSH) which is a part of the CEBS. The empirical results suggest that clustering normal vectors may result in near collinear cluster centers and boundaries between clusters of similar trend, thus pointing to axis of a potential megafold. Clustering dip vectors resulted on the other hand in near co-circular cluster centers, thus pointing to a potential megacone. We also show that the linear arrangements of the anomalies, their topological relationships and internal structure can provide insights regarding the internal structure of the singularity, e.g. whether it may be due to drilling a nonvertical fault plane or due to a wider deformation zone composed of many smaller faults.



# 1 Introduction

## 1.1 Detecting three-dimensional outliers

Current workflows in 3D geological modeling lack the capacity to examine 3D geometric outliers from datasets collected throughout geological terrains, likely leaving structural information undiscovered. When considering structural attributes obtained from 3D seismics, maps showing spatial distribution of dip angle or dip direction can be helpful to detect structures (Roberts, 2001; Di and Gao, 2017). For example, if a preferred dip direction of strata exists, then setting threshold to dip angle

may be used to detect faults striking perpendicular to the preferred dip direction. However, neither of these attributes is three-dimensional in nature: for example, dip angle is not capable of showing the dip direction of faults and vice-versa. Moreover, these methods are not decisive in terms of detecting rare observations, i.e. observations that differ significantly from the majority of the data. In other words, in these 1D or 2D approaches the boundaries between values of dip angle or dip direction are often established without any optimization criterion (e.g. when using available default color palettes) which may result in

the lack of spatial integrity of potential structures and following difficulties in their identification.

## 1.2 Optimization and clustering

Machine learning is a promising tool for anomaly (outlier) detection. In supervised approaches, observations must be labeled which sometimes involves subjectivity (Bergen et al., 2019). From the definition of supervised approaches, it follows that they are not specifically designed to explore potentially new patterns of data. In contrast, in unsupervised techniques, distance

metrics can be used to identify sets of similar observations (Hastie et al., 2009). In clustering algorithms, the objective of finding homogenous subsets is often realized by optimization: minimizing the within-cluster dissimilarity or, equivalently, by maximizing the between-cluster similarity.

## 1.3 Study Introduction

In this paper, we propose clustering the 3D terrain observations using cosine distance as dissimilarity metric. This goal is

achieved using a fact that squared Euclidean distance applied for unit vectors is proportional to the cosine distance (Zhang et al., 2020; Choi et al., 2014) (see also Methods). While (Michalak et al., 2019) used only normal vector representation for extracting orientation statistics, we illustrate the cartographic differences for two 3D representations of the studied terrains: normal and dip vectors with two versions of spatial clustering: irregular and regular (see also Methods).

# 2 State of the art

## 2.1 Subjectivity

Geology is considered to be a subjective science (Curtis, 2012). For example, in geological maps showing spatial distribution of dip direction, the boundaries between colors representing dip direction domains may result from using predefined color





palettes available in GIS software (Cawood et al., 2017). This poses the risk that in these simple methods, the boundaries between domains have no geometric meaning: no optimization techniques were used to group geometrically similar observations in one color domain. In the field of subsurface structural geology, it is expected that the recognition of subjectivity will bring about a better understanding of the subsurface and can help in reducing uncertainties (Bond, 2015; Curtis, 2012). Having a method to detect and investigate the spatial configurations of 3D geometric outliers (grouped in homogenous subsets according to optimization criterion) and their topology is expected to be a major step in the understanding of the subsurface architecture. This is in particular valid for environments with sparse data, in which the topology of the fault network may be uncertain or prone to the existence of unknown faults (Schneeberger et al., 2017).

## 2.2 Examples of using machine-learning in solid Earth geoscience

In fields related to solid Earth geosciences, machine-learning methods have been applied in earthquake prediction (Seydoux et al., 2020; Johnson et al., 2021; McLellan and Audet, 2020), the classification of rock units in geological mapping based on lithological or geophysical features (Cracknell and Reading, 2014; Kuhn et al., 2018; Xiong and Zuo, 2021; Wang et al., 2020) and the investigation of the topology of fractured networks (Srinivasan et al., 2018; Valera et al., 2018). Unsupervised techniques have also been successfully used for the problem of finding homogenous subsets of observations representing discontinuities (Hammah and Curran, 1999; Zhan et al., 2017a, b) or portions of geological interfaces to determine the average orientation of regional trends (Michalak et al., 2019). The above studies did not however investigate the role of vector representations (normal and dip vectors) on the clustering results. They also did not attempt to use the characterization of Voronoi diagrams to explain the meaning of the boundaries between obtained clusters.

## 2.3 Spatial clustering

Spatial clustering is a generic term for investigating geometric trends throughout a surface of interest (Fisher, 1993; Fisher et al., 1985). In the first step of the procedure, orientations with spatial components are sampled throughout the surface. Then, the observations are grouped into homogenous subsets. The partition can be achieved by assigning a corresponding class resulting from clustering. This class can be recorded as an integer and then represented with a label (a symbol or a color). From a technical viewpoint, the data frame used for clustering is reduced because only coordinates of normal and dip vectors serve as input for clustering. Thus, the spatial distribution of these classes can be examined by re-assigning a spatial component to the labeled observations. The idea of spatial clustering shares some similarities with color-coding portions of interpreted seismic surfaces with respect to their orientation (Di and Gao, 2017). However, the latter approach does not employ optimization techniques (see section 2.1 Subjectivity) to group individual observations into homogenous subsets. Moreover, color coding is usually performed independently for dip and dip direction (Di and Gao, 2017; Roberts, 2001), which does not allow investigation of the relationship between dip and dip direction anomalies.



## 3 Methods

### 3.1 Calculating the orientation


Three noncollinear points in three-dimensional space define a plane whose orientation can be calculated using basic linear algebra (Allmendinger, 2020; Groshong, 2006). When triangulation of the points is applied, the quality of the triangles can be measured using a variety of nonequivalent coefficients for their further removal (Collon et al., 2015; Michalak, 2018; Frey and Borouchaki, 1999). In this study, we used a collinearity coefficient defined as the proportion of the longest triangle edge to the

length sum of the remaining edges (Michalak, 2018). The resulting coefficient lies within the interval of [0.5, 1], with lower values pointing at equilateral triangles and higher values representing collinear configurations. In the irregular grid (which is the case for the KSH), we filtered the configurations whose collinearity exceeded 0.90. This restriction resulted primarily in the removal of triangles that lie at the edge of the convex hull (e.g., see the nonconvex shape of the convex hull in Fig.14).

### 3.2 Assumptions underlying the clustering procedure

Selected clustering algorithms allow dissimilarity metrics to be specified to evaluate the similarity (or dissimilarity) between individual observations. In our case, these observations are normal and dip vectors (Fig. 1). For example, if the k-means algorithm is used (James et al., 2013), the squared Euclidean distance acts as the distance metric between p-dimensional observations $x_i$ and $x_{i'}$ (in our case $p = 3$):

$$d(x_i, x_{i'}) = \sum_{j=1}^{p} \left(x_{ij} - x_{i'j}\right)^2 = \left|\left|x_i - x_{i'}\right|\right|^2 \qquad Eq.(1)$$

We propose to use a well-known fact (Choi et al., 2014; Zhang et al., 2020) that for unit normal vectors, the above squared distance is proportional to cosine distance ("∘" denotes the scalar product between two vectors, and $||\cdot||$ is the Euclidean norm):

$$\sum_{i=1}^{n}\sum_{j=1}^{n}\left|\left|x_i - x_{i'}\right|\right|^2 = \sum_{i=1}^{n}\sum_{j=1}^{n}\left(\left|\left|x_i\right|\right|^2 - 2x_i{}^{\circ}x_{i'} + \left|\left|x_{i'}\right|\right|^2\right) \qquad Eq.(2)$$

$$= \sum_{i=1}^{n}\sum_{j=1}^{n}(1 - 2x_i{}^{\circ}x_{i'} + 1) \qquad Eq.(3)$$

$$= \sum_{i=1}^{n}\sum_{j=1}^{n}(2 - 2x_i{}^{\circ}x_{i'}) \qquad Eq.(4)$$


$$= \sum_{i=1}^{n}\sum_{j=1}^{n}2(1 - x_i{}^{\circ}x_{i'}) = 2\sum_{i=1}^{n}\sum_{j=1}^{n}(1 - x_i{}^{\circ}x_{i'}) = 2\sum_{i=1}^{n}\sum_{j=1}^{n}1 - \frac{x_i{}^{\circ}x_{i'}}{1 * 1} \qquad Eq.(5)$$

$$= 2\sum_{i=1}^{n}\sum_{j=1}^{n}1 - \frac{x_i{}^{\circ}x_{i'}}{||x_i|| * ||x_{i'}||} \stackrel{\text{def}}{=} 2\sum_{i=1}^{n}\sum_{j=1}^{n}1 - \cos\left(\sphericalangle\{x_i, x_{i'}\}\right) . \; Eq.(6)$$



Thus, the optimization problem solved in the k-means algorithm for unit vectors can be conceptualized in two ways (using the fact the $cos(x)$ takes values on the interval $[-1,1]$):

A.  Minimizing the within-cluster cosine distance. $1 - \cos(\sphericalangle\{x_i, x_{i'}\})$ is minimized (equals zero) when $\cos(\sphericalangle\{x_i, x_{i'}\}) = 1$, i.e. when the angle between $x_i$ and $x_{i'}$ is 0°.

B.  Maximizing the between-cluster cosine distance. $1 - \cos(\sphericalangle\{x_i, x_{i'}\})$ is maximized (equals two) when $\cos(\sphericalangle\{x_i, x_{i'}\}) = -1$, i.e. when the angle between $x_i$ and $x_{i'}$ is 180°.

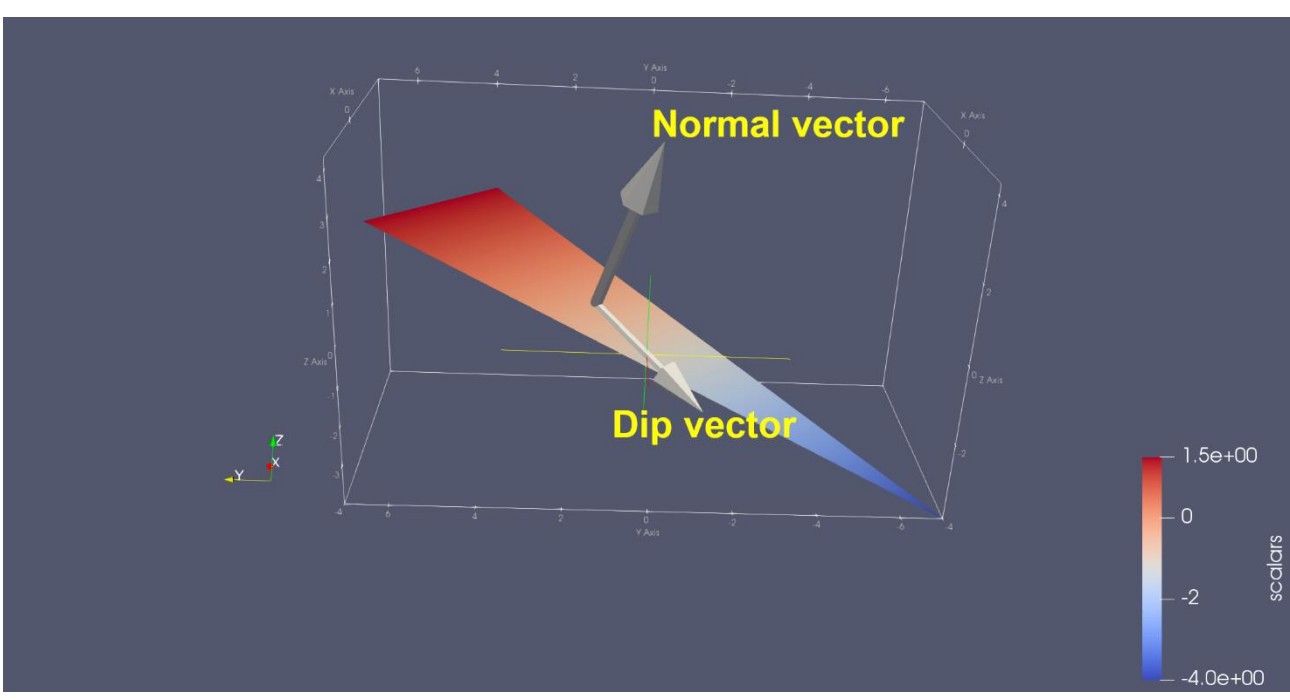

**Figure 1.** A triangle with its normal and dip vectors. These two representations are used in our study for clustering.

### 3.3 Equivalence of k-means clustering and Voronoi cells

Because the clusters resulting from applying the k-means algorithm can be conceptualized as Voronoi cells (Hastie et al., 2009), we propose to explain some of the clustering results using relevant computational geometry theorems about Voronoi diagrams (De Berg et al., 2008).

Theorem 1. *Let P be a set of n point sites in the plane. If all the sites are collinear then* Vor(P) *consists of n−1 parallel lines. Otherwise,* Vor(P) *is connected and its edges are either segments or half-lines.*

Theorem 2. *For the Voronoi diagram* Vor(P) *of a set of points P the following holds:*





*(i) A point q is a vertex of* Vor(*P*) *if and only if its largest empty circle CP(q) contains three or more sites on its boundary.*

*(ii) The bisector between sites pi and pj defines an edge of* Vor(*P*) *if and only if there is a point q on the bisector such that CP(q) contains both pi and pj on its boundary but no other site.*

### 3.4 Irregular and regular trend maps

We propose here two versions of spatial clustering of geological contacts. The first version is based on taking one

representative associated with the interiors of Delaunay triangles (their geometric centers). The vertices corresponding to the Delaunay triangles are given indices that are also included in the resulting data frame, which we denote as **Table Y**. The clustering methods assign the resulting orientation labels to the geometric centers of the Delaunay triangles (Fig. 2).

The second version tries to evenly represent the geometric trends throughout the deformed surface (Fig. 2). It allows to answer whether a specific a 2D point $p$ (a geographic location) lying in the domain of a triangulation $T$ is affected by a 3D geometric

anomaly related to a triangulation model $T$. In this approach, a structured grid defined by points linked with corresponding clustering labels is generated. From a technical viewpoint, it is first necessary to generate the orientations of the Delaunay triangles. Then, we generate a regular point network within a region of interest. Next, we use the *locate* function offered by the CGAL library to link the points from the regular network with the corresponding triangles. In its simplest version, this function takes the point **query** as its argument, and the possible return values are as follows (Yvinec, 2021):

- If the point (from the regular network) **query** lies inside the convex hull of the points (boreholes), a face that contains the **query** in its interior or on its boundary is returned.

- If the point **query** lies outside the convex hull of the triangulation but in the affine hull, it is a face ($\infty$, p, q) such that the **query** lies to the left of the oriented line (the rest of the triangulation lies to the right of this line).

In the code, we enable only the inclusion of the first group of triplets into the resulting data frame, which we denote here as

**Table X**. To ultimately link the points from the regular grid (included in **Table X**) with the orientation labels (**Table Y**), the corresponding data frames need to be merged. Note that **Table Y** includes all available observations, but this is not the case for **Table X**. This is because not all triangles (e.g., those of small size) may be linked with any points from the user-defined regular grid, and this can especially be the case if the grid is sparse. To observe the differences, we therefore recommend using a SQL-related **Right Outer Join (ROJ)** (or Left Outer Join with replaced arguments) method rather than **Inner Join** for

merging. This choice allows us to calculate the proportion of the area that is not covered with any orientation label (the coverage presented in Fig. 14). **ROJ** applied to Tables **X** and **Y** returns all rows from the right Table (**Y**) and any rows with matching keys from the left Table (**X**). In this study, the keys over which the tables are merged are the indices of the boreholes that build the corresponding Delaunay triangles. If no points can be found for a given triangle, **NA** values are assigned to the *px* and *py* coordinates in the merged **Table Z**.



**Figure 2**. Two versions of spatial clustering conducted on a geological contact: (A) geometric centers corresponding to Delaunay triangles were selected; (B) regular grid based on CGAL queries and merging techniques; (C) orientation partitioning with respect to which the trends are visualized; (D) a scheme presenting the role of CGAL (CGAL::locate) queries and merging (right outer join) techniques in obtaining the final result



# 4 Geological setting

## 4.1 Case study 1: The Central European Basin System

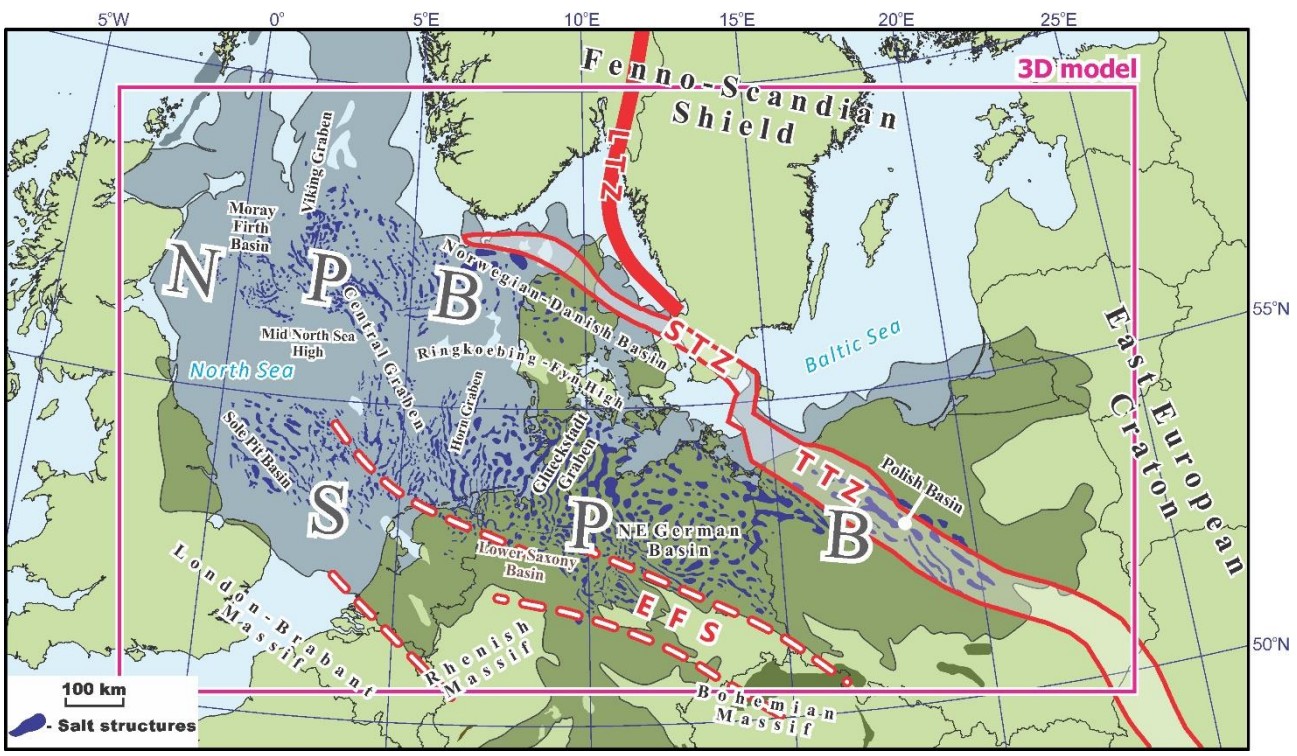

**Figure 3.** Tectonic settings within the Central European Basin System with the location of the regional-scale 3D structural
model (Maystrenko and Scheck-Wenderoth, 2013; Maystrenko et al., 2013, 2012). LTZ is according to (Medhus et al., 2012).
Abbreviations: EFS — Elbe Fault System, LTZ — Lithospheric Transition Zone, NPB — Northern Permian Basin, SPB —
Southern Permian Basin, STZ — Sorgenfrei–Tornquist Zone and TTZ — Teisseyre-Tornquist Zone.

The sedimentary cover of the CEBS can be subdivided into two clearly distinguished structural levels: (1) pre-Permian, (2)
Permian and (3) Meso–Cenozoic structural units (Doornenbal et al., 2009; Evans et al., 2003; Maystrenko et al., 2013, 2012;
Ziegler, 1990b). The pre-Permian sedimentary level mainly includes Devonian and Carboniferous sedimentary rocks with
Silurian, Ordovician and Cambrian rocks along the north-eastern margin of the CEBS (Doornenbal et al., 2009; Evans et al.,
2003; Fossen et al., 2017; Lassen and Thybo, 2012; Usaityte, 2000; Wiest et al., 2020).

In the Late Carboniferous–Early Permian, the CEBS area was affected by the regional-scale rifting with deposition of mainly
clastic sediments within the Northern and Southern Permian basins (Abramovitz and Thybo, 2000; Benek et al., 1996; Dadlez
et al., 1995; Heeremans et al., 2004; Maystrenko et al., 2008; Plein, 1990; Stemmerik et al., 2000; Ziegler, 1990b). During the



Middle-Late Permian time, a large amount of Zechstein rock salt, anhydrite and carbonates accumulated within the Permian basins (Geluk, 2000; Maystrenko et al., 2008; Plein, 1990; Ziegler, 1990b). The Zechstein salt was reactivated during the Mesozoic and Cenozoic to form various salt structures within the CEBS (Fig. 3) (Doornenbal et al., 2009; Evans et al., 2003; Maystrenko et al., 2012, 2013).

Several pulses of active tectonics characterized the post-Permian evolution of the CEBS, including Triassic extension, Late Jurassic-Early Cretaceous extension/transtension and Late Cretaceous-Early Cenozoic compression/inversion (Bell et al., 2014; Dadlez et al., 1995; Doornenbal et al., 2009; Erratt et al., 1999; Evans et al., 2003; Frederiksen et al., 2001; Graversen, 2002; Kyrkjebø et al., 2004; Mazur et al., 2016; Odinsen et al., 2000; Phillips et al., 2019; Scheck-Wenderoth and Lamarche, 2005; Vejbæk and Andersen, 2002; Ziegler, 1990b).

The post-Permian tectonic events led to the formation of local sub-basins, superimposed on the Northern and Southern Permian basins (Figs. 3, 4). The Norwegian-Danish Basin, the Central, the Viking and the Horn grabens were formed within the Northern Permian Basin during the Triassic and the Jurassic-Early Cretaceous (Baldschuhn et al., 2001; Clausen and Korstgård, 1996; Erratt et al., 1999; Møller and Rasmussen, 2003; Phillips et al., 2019; Vejbæk, 1990). In the Southern Permian Basin, the Glueckstadt Graben together with the Northeast German and the Lower Saxony basins formed the broad North

German Basin (Baldschuhn et al., 2001; Betz et al., 1987; Brink et al., 1990; Kockel, 2002; Maystrenko et al., 2005; Scheck et al., 2003) as well as the Polish Basin was superimposed on the eastern margin of the CEBS (Dadlez, 2003; Krzywiec, 2006). During the Late Cretaceous-Early Cenozoic, some parts of the CEBS were undergone to basin inversion with the strongest compressive deformations and uplift along the Elbe Fault System, the Sorgenfrei-Tornquist and the Teisseyre-Tornquist zones (Hansen et al., 2000; Krzywiec et al., 2021; Mazur et al., 2005; Mogensen and Jensen, 1994; Otto, 2003; Scheck-Wenderoth and Lamarche, 2005; Voigt et al., 2008; Ziegler, 1990a). During the Cenozoic, broad subsidence occurred within the central

part of the North Sea, where more than 3 km of sediments were deposited (Evans et al., 2003; Maystrenko et al., 2013; Ziegler, 1990b).







**Figure 4.** A: Hypsometry of the bottom Jurassic surface from the Central European Basin System. B: Dip angle calculated for
the bottom Jurassic surface. By definition, the dip direction of potential anomalies cannot be observed. C: Dip direction
calculated for the Jurassic horizon. By definition, the magnitude of slopes is not visible. The boundaries between colors (A,
B) are established without considering an optimization criterion.

## 4.2 Case study 2: The Kraków Silesian Homocline (KSH)

The KSH in Poland is interpreted as an eastern continuation of a greater geological unit called the Fore-Sudetic homocline
(Fig. 5) (Stupnicka and Stempień-Sałek, 2016; Mizerski, 2020; Narkiewicz, 2020). Both units are interpreted as natural slopes
of the Szczecin-Łódź-Miechów synclinorium formed during Late Cretaceous-early Paleocene inversion of the Permian-
Mesosoic Polish basin (e.g. Dadlez et al., 1995; Słonka and Krzywiec, 2019). The axial part of the Permian-Mesosoic Polish
basin is called the mid-Polish trough and strikes parallel to the TTZ (see Mazur et al., 2021 for the latest interpretation of TTZ).
The mid-Polish trough was a zone of strong subsidence during the late Permian and Mesozoic (Kutek and Głazek, 1972). The
inversion and uplift of the Polish basin resulted in the formation of the Mid-Polish anticlinorium, with two synclinoria
symmetrically distributed along the NW-SE trending anticlinorium (Fig. 5, Kutek and Głazek, 1972; Narkiewicz, 2020).
The Permo-Mesozoic deposits of the KSH represent predominantly clastic and carbonate series with common hiatuses and lie
unconformably on the denuded and morphologically diversified Paleozoic or Precambrian basement (Buła et al., 2015). It is
generally assumed that the strata along with accompanying geological contacts dip gently toward the NE, a feature that has
been present since the late Cimmerian phase (Górecka, 1993; Krokowski, 1984). The layers were ultimately tilted to the NE
(Figs. 5, 6), in the direction of the Szczecin-Łódź-Miechów synclinorium axis, during the inversion of the mid-Polish trough
in the Maastrichtian and Paleocene (Górecka, 1993; Kutek and Głazek, 1972). Deviations from the preferred direction of the
dip to the NE can be observed in the southeastern part of the KSH, where layers may dip to the S (Krokowski, 1984). The
latter effect is believed to be caused by the thrusting Carpathians and the corresponding bending of deposits north of the
thrusting loads in the Miocene (Jarosiński et al., 2009; Krokowski, 1984). Additionally, locally greater angular dip angles are
expected to be found near faults due to fault-related bending of strata (Bednarek et al., 1992; Matyszkiewicz and Krajewski,
1996; Krokowski, 1984). It is important to note that some of the observed or hypothesized faults have received kinematic
interpretations. For example, the en echelon arrangement of SW-NE trending faults (Fig. 5) is believed to be formed by the
underlying dextral movement of a strike-slip fault (Krokowski, 1984). This hypothesized movement, trending SE-NW, is
believed to be related to the parallel Kraków-Lubliniec Fault, which separates blocks of crystalline basement (Buła et al.,
2015).
For this study, we selected the geological interface that separates younger ore-bearing clays (late Bajocian-late Bathonian)
from the older Kościeliska sandstones (early Bajocian) (Kopik, 1998) in the area of Nowa Wieś town. There is a hiatus between
the sediments, evidenced by the lack of a Strenoceras subfurcatum ammonite zone (Zakrzewski, 1976).



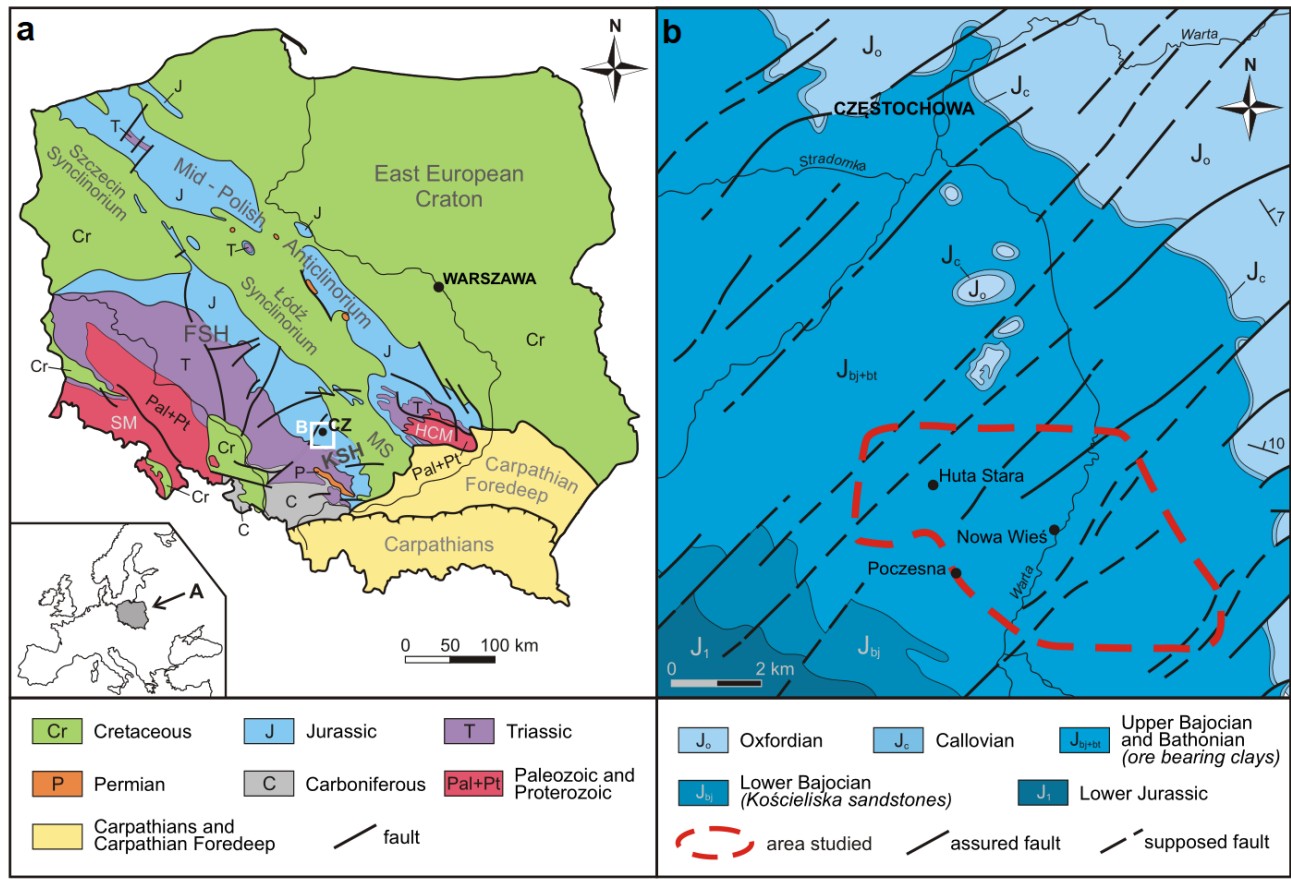

**Figure 5.** A - Simplified geological map of Poland without Cenozoic formations (modified after Karnkowski, 2008; Osika et al., 1972); and the location of area studied. CZ - Częstochowa, FSH - Fore-Sudetic Homocline, HCS - Holy Cross Mountains, KSH - Kraków-Silesian Homocline, MS - Miechów Synclinorium, SM - Sudety Mountains. B - Geological map of the studied part of the Kraków-Silesian Homocline (modified after Bardziński et al., 1986) and the location of the study area.









**Figure 6.** A: Hypsometry of the Jurassic horizon in Kraków-Silesian Homocline. B: Dip angle calculated for the Jurassic horizon. By definition, the dip direction of potential anomalies cannot be observed. C: Dip direction calculated for the Jurassic horizon. By definition, the magnitude of slopes is not visible. The boundaries between colors are established without
considering an optimization criterion which results in the lack of spatial integrity of potential structures.

## 5 Materials

### 5.1 Case study 1 (CEBS) - materials

We used the lithosphere-scale 3D structural model of the CEBS, constructed by Maystrenko et al. and Maystrenko and Scheck-Wenderoth (Maystrenko et al., 2013, 2012; Maystrenko and Scheck-Wenderoth, 2013), as the primary structural skeleton. The
horizontal resolution of the model (grid spacing) is 4000 m. Vertically, the 3D model consists of 17 layers, nine of which are the following sedimentary layers: Cenozoic, Cretaceous, Jurassic, Triassic, Permian salt, Permian carbonates, Rotliegend sediments, Permo-Carboniferous volcanics and pre-Permian sedimentary rocks. In addition, the 3D structural model of the CEBS includes six layers of the crystalline crust and one layer for the lithospheric mantle. All depth interfaces, such as structural bases of sedimentary layers, top of the crystalline basement, Moho and lithosphere-asthenosphere boundary, can be
extracted from the model. In this study, a base of the Jurassic has been taken to analyse the structural features of the CEBS. The lithosphere-scale 3D structural model covers the entire Northern and Southern Permian basins (Fig. 3). The adjacent London-Brabant, Rhenish and Bohemian massifs in the south and the Fenno-Scandian Shield and the East European Craton in the north-east are also partially covered by the model. The constructed model of the CEBS is based on the available structural data, such as boreholes, seismic data and maps. A detailed description of the input data is given in Maystrenko et al.
(Maystrenko et al., 2012, 2013, 2020). Here, we will only mention the largest data sets used during the model construction. The main data source for the 3D model was the North Sea Digital Atlas which covers the entire North Sea (PGS, 2003). The digital version of the Geological Atlas of the Netherlands has been used for the Netherlands (NITG, 2004). To cover the North German Basin, the Geotectonic Atlas of northwest Germany (Baldschuhn et al., 2001), the 3D structural model of the Glueckstadt Graben (Maystrenko et al., 2006) and the 3D model of the NE German Basin (Scheck et al., 2003) have been
taken. The 3D crustal-scale model of the Polish Basin (Lamarche and Scheck-Wenderoth, 2005) has been used for Poland.

### 5.2 Case study 2 (KSH) - materials

We used 810 borehole records (Anon, The borehole database…) that were handed over to the University of Silesia in Katowice by the "Geological Company of Częstochowa" (Częstochowskie Przedsiębiorstwo Geologiczne). The digitized version of these records contains Cartesian coordinates for the studied interface. The uncertainty of the borehole paths was not provided, and the precision of the coordinates was 1 cm. We used Pulkovo 1942(58)/Poland zone V (EPSG: 2175) as the coordinate
reference system. Compared to a previous study built upon this dataset (Michalak et al. 2019), we filtered the data to minimize



the number of "noisy" boreholes that are most likely related to measurement errors. The assumed errors were manifested as unusual pointwise distributed depressions or elevations of the studied surface.

## 6 Results

### 6.1 Determining the optimum number of clusters

An important clustering issue is the selection of the number of clusters. There are many competitive heuristics suggesting the optimal number of clusters (Rousseeuw, 1987; Hastie et al., 2009). Since in this study we only use the k-means algorithm, we followed the idea of the elbow method. It requires to run the k-means algorithm for different k (e.g. from 1 to 10) and to compute the total within-sum of squares for each clustering. To locate the optimal number of clusters, one looks for a kink in the sum of squares curve (Hastie et al., 2009). The experimental results usually suggest the optimum number of clusters to be 2 (Fig. 7B, 7C), 3 (Fig. 7A, 7D) or 4 (Fig. 7B).



**Figure 7**. The elbow method to determine the optimal number of clusters. It was applied to different case studies: (A) CEBS with normal vector representation, (B) CEBS with dip vector representation, (C) KSH with normal vector representation, (D) KSH with dip vector representation.

## 6.2 Case study 1 (CEBS) - results

The linear, anomalous zones are obtained along the NW-SE trending lithosphere-scale fault zones, such as the Tornquist Zone (the Sorgenfrei-Tornquist and the Teisseyre-Tornquist zones) in the north-east and the Elbe Fault System in the south-west (Figs. 4, 8-10). This is due to the fact that these fault zones controlled the structure of the crystalline basement and the configuration of the sedimentary cover within the CEBS that is reflected by the spatial clustering in Figs. 10A and 10B. In particular, the structural development of the Permo-Mesozoic Polish Basin is strongly coupled with the Teisseyre-Tornquist Zone, which can be considered as the preexisting lithospheric weak zone beneath this basin (Mazur et al., 2021). The Teisseyre-



Tornquist Zone is characterized by two wide zones of low angle dip direction (Fig. 4C, 10B), reflecting the NW-SE strike of
the Polish Basin (Fig. 5). In the case of the Sorgenfrei-Tornquist Zone, the north-eastern margin of the Norwegian-Danish
Basin was significantly affected by the shape of this fault zone during both the Permo-Mesozoic subsidence and the Late
Cretaceous uplift (Erlström et al., 1997). A similar situation is along the Elbe Fault System which is also represented by a zone
of weakness at the south-western margin of the North German Basin (Scheck et al., 2002). Another area with a set of the linear
anomalous zones is located within the North Sea (Figs. 4, and 10B), where the Moray Firth Basin, the Central and Viking
grabens are located (Fig. 3). There, these linear anomalous zones are mainly caused by the major boundary faults of the graben
structures with the mentioned sedimentary basins. The sedimentary cover is faulted and folded along the boundary faults and,
therefore, the geometric anomalies follow the configuration of the faults.

The anomalous zones with smaller linear size and higher dip angles are most pronounced within the North German Basin, the
Central Graben, the Horn Graben and the eastern part of the Norwegian-Danish Basin where the sedimentary cover is pierced
and strongly deformed by large salt structures (Figs. 4 and 10A). The largest anomalies are obtained within the Glueckstad
Graben and outline the NE-SW trending, long and wide salt walls located there. Actually, the large size of the anomalies and
high dip angles correlate well with the high intensity of salt movements, which were the strongest within the Glueckstad
Graben (Maystrenko et al., 2013; Trusheim, 1960; Warsitzka et al., 2019). The high dip angle of the small-scale anomalies is
obtained within the northern part of the Central Graben, where the salt diapirs pierce and strongly deform the sedimentary
layers in the vicinity of salt structures (Davison et al., 2000; Karlo et al., 2014; Rank-Friend and Elders, 2004). The salt
movements were also intensive within the Polish Basin (Krzywiec, 2004, 2012). However, the salt-induced deformations of
the sedimentary cover of the Polish Basin are not clearly reflected by spatial clustering analysis compared to the Glueckstadt
Graben or Central Graben. This is mostly related to a relatively low resolution of the data which have been used to construct
the 3D structural model of Poland (Lamarche and Scheck-Wenderoth, 2005). In contrast, the input data for the Glueckstadt
and Central grabens were characterized by higher resolution (Baldschuhn et al., 2001; PGS, 2003), allowing authors to include
more details of the basin structures in the grabens.



**a  Clustering normal vectors**

**b  Clustering dip vectors**



**Figure 8.** Using k-means (k=2) clustering to normal (A) and dip (B) vectors for the investigated CEBS Jurassic horizon (irregular versions). We used 236380 observations for clustering but the visualization is based on a random sample (10 000 observations). It can be observed that the normal representation (A) generated two sets of clusters with the less represented (about 3%) magenta cluster dipping at moderate angles to SW. Clustering dip vectors (B) resulted in partitions that represent two dip direction domains with two cluster centers having similar dip angles. Dip and dip direction of cluster centers are given in Tab. 1.



## a  Clustering normal vectors

## b  Clustering dip vectors




**Figure 9.** Using k-means (k=3) clustering to normal (A) and dip (B) vectors for the investigated CEBS Jurassic horizon (irregular version). We used 236380 observations for clustering but the visualization is based on a random sample (10 000 observations). It can be observed that the normal representation (A) generated two sets of clusters with the less represented (both less than 3%) magenta and green clusters dipping at moderate angles to SW and NE, respectively. Clustering dip vectors (B) resulted in partitions that represent three dip direction domains (NE, W and SSE) with a common vertex near the stereonet
origin and cluster centers having similar dip angles. Dip and dip direction of cluster centers are given in Tab. 1.



## a  Clustering normal vectors

## b  Clustering dip vectors



**Figure 10.** Using k-means (k=4) clustering to normal (A) and dip (B) vectors for the investigated Jurassic horizon (irregular version). We used 236380 observations for clustering but the visualization is based on a random sample (10 000 observations). It can be observed that the normal representation (A) generated almost collinear cluster centers (NW-SE). Clustering dip vectors (B) resulted in partitions that represent four dip direction domains: NNE (black), W (magenta), E (green) and SSW (blue) with a common vertex near the stereonet origin and cluster centers having similar dip angles. Dip and dip direction of cluster centers are given in Tab. 1.

**Table 1**. Dip and dip direction of cluster centers (CEBS)

| Center name | Dip angle | Dip direction |
|---|---|---|
| Two clusters: 1st center (normal) | 0,31 | 44,91 |
| Two clusters: 2nd center (normal) | 8,97 | 225,23 |
| Three clusters: 1st center (normal) | 9,37 | 47,80 |
| Three clusters: 2nd center (normal) | 0,03 | 12,76 |
| Three clusters: 3rd center (normal) | 9,68 | 225,02 |
| Four clusters: 1st center (normal) | 4,35 | 36,74 |
| Four clusters: 2nd center (normal) | 0,15 | 218,75 |
| Four clusters: 3rd center (normal) | 18,14 | 58,78 |
| Four clusters: 4th center (normal) | 10,13 | 224,96 |
| Two clusters: 1st center (dip) | 2,34 | 48,45 |
| Two clusters: 2nd center (dip) | 2,27 | 228,81 |
| Three clusters: 1st center (dip) | 1,76 | 164,61 |
| Three clusters: 2nd center (dip) | 1,86 | 263,86 |
| Three clusters: 3rd center (dip) | 2,13 | 40,64 |
| Four clusters: 1st center (dip) | 1,89 | 9,30 |
| Four clusters: 2nd center (dip) | 1,74 | 87,16 |
| Four clusters: 3rd center (dip) | 1,81 | 187,69 |
| Four clusters: 4th center (dip) | 1,73 | 265,79 |

## 6.3 Case study 2 (KSH) - results

Both the normal (Figs.11A, 12A, 13A) and dip vector representations (Figs.11B, 12B, 13B) reveal similar spatial configurations of geometric anomalies, i.e., observations dissimilar to the subhorizontal dip to the NE. A visible difference between normal and dip vector representation can be attributed to the spatial integrity of W-E trending anomalies (dipping to S) which are not very well preserved in the normal vector representation with four clusters. This effect can be explained by



the fact that in the normal representation with four clusters  the cluster centers are more or less collinear. This suggest (Theorem 1) that in the normal vector representation boundaries between clusters are more or less parallel. Indeed, in our results the boundaries between clusters seem to have a similar NE-SW trend (Fig. 13A). The implication of this result is that observations dipping to S may be geometrically far from the cluster center (SE) and thus may be assigned to more than one cluster (the green one and the black one in this case).

For the two representations, two distinctly trending sets can be observed: NE-SW and NNE-SSW (locally N-S). The presence of an en echelon arrangement of the NE-SW set is in line with the orientation of faults in the Częstochowa region (Fig. 5B - Bardziński et al., 1986) and with the model of extensional faulting in the northern part of the KSH due to SE-NW-oriented dextral strike-slip movement (Krokowski, 1984). However, our results do not support the unimodal distribution of fault strikes (only NE-SW), as proposed by (Bardziński et al., 1986, Fig. 5B).

We see that some of the arrangements are composed of more than one observation in the direction perpendicular to their trend (e.g., the SW-NE trending anomaly in the NW part of the area in Figs.11B). This necessitates discussion about the origin of these forms given that the difference in elevation between a hanging wall and footwall can be consumed by a single triangle (e.g., with two points lying on the hanging wall and the third on the footwall) (Michalak et al., 2021).

The above effect could be explained by several competitive hypotheses. For example, the fault plane could have been drilled,
thus broadening the zone of triangles genetically related to the fault (Michalak et al., 2021). Assuming the tectonic origin of the related structures, it can be hypothesized that fault drags on the hanging wall contribute to subsidiary elevation differences that must be consumed by nearby triangles. It could also be argued that an unusual lowering of the contact surface is due to a deformation zone composed of many smaller faults. Another hypothesis could be that the related feature is not a fault but rather a sedimentary slope, which would explain the gradual lowering of the contact surface.

From a topological (Thiele et al., 2016) perspective, some of the NE-SW trending arrangements are paired in that their NW- and SE-dipping counterparts are adjacent (e.g., the form composed of blue and magenta SW-NE-trending belts in the S part of the area in Fig. 14C, D). Depending on their relative position, they can be interpreted either as paleovalleys, grabens and synclines (negative forms) or peaks, horsts and anticlines (positive forms). They can also be interpreted in terms of antithetic shear with hanging walls dipping against the main fault, which is often the case for listric faults (Harding and Tuminas, 1989; 375     Fossen, 2006) or reverse drag/rollover anticlines (Fossen, 2006).

We also produced results using the regular version of spatial clustering with different coverage rates (Figs. 14A-D), i.e. proportions of triangles linked with points comprising the regular grid. These results show that lower coverage may result in the omission of small triangles (Fig. 14A, 14B) which can be misleading for analysis of connectivity between different representatives of the observed anomalies (Fig. 14C, 14D). The regular version allows to better observe the orientation of 380     neighbors of triangles which can be more difficult if the irregular version is applied.





**a  Clustering normal vectors**



**b  Clustering dip vectors**



**Figure 11.** Using k-means (k=2) clustering to normal (A) and dip (B) vectors for the investigated KSH Jurassic horizon. This version is irregular: orientation labels are assigned to geometric centers of Delaunay triangles. Both clustering and visualizations are based on 1502 observations. It can be observed that the normal representation (A) generated two sets of clusters with the less represented (about 4.7 %) magenta cluster dipping at moderate angles to NW. Clustering dip vectors (B) resulted in partitions that represent two dip direction domains with NW and ENE centers having similar dip angles. Dip and dip direction of cluster centers are given in Tab. 2.



## a Clustering normal vectors

## b Clustering dip vectors





**Figure 12.** Using k-means (k=3) clustering to normal (A) and dip (B) vectors for the investigated KSH Jurassic horizon. This
version is irregular: orientation labels are assigned to geometric centers of Delaunay triangles. Both clustering and
visualizations are based on 1502 observations. It can be observed that the normal representation (A) generated three sets of
clusters with the less represented (about 10.7 %) magenta and blue (about 4.5%) clusters dipping to NW and SE, respectively.
The boundaries between clusters have a similar NE-SW trend. Clustering dip vectors (B) resulted in partitions that represent
three dip direction domains with NW, NE and ESE centers having similar dip angles. Dip and dip direction of cluster centers
are given in Tab. 2.





**a Clustering normal vectors**

**b Clustering dip vectors**



**Figure 13.** Using k-means (k=4) clustering to normal (A) and dip (B) vectors for the investigated KSH Jurassic horizon. This version is irregular: orientation labels are assigned to geometric centers of Delaunay triangles. Both clustering and visualizations are based on 1502 observations. It can be observed that the normal representation (A) generated four sets of clusters with the less represented magenta (about 15.4 %), black (about 5.9 %) and blue (about 3.3 %) clusters dipping to NW (at small angles), NW (at moderate angles) and to ESE (at moderate angles), respectively. The boundaries between clusters have a similar NE-SW trend. Clustering dip vectors (B) resulted in partitions that represent four dip direction domains with NW, NE and ESE centers having similar dip angles. Dip and dip direction of cluster centers are given in Tab. 2.

**Table 2.** Dip and dip direction of cluster centers (KSH)

| Center name | Dip angle | Dip direction |
|---|---|---|
| Two clusters: 1st center (normal) | 1,11 | 61,77 |
| Two clusters: 2nd center (normal) | 13,31 | 317,80 |
| Three clusters: 1st center (normal) | 5,27 | 112,64 |
| Three clusters: 2nd center (normal) | 0,99 | 29,08 |
| Three clusters: 3rd center (normal) | 13,70 | 317,66 |
| Four clusters: 1st center (normal) | 1,22 | 65,32 |
| Four clusters: 2nd center (normal) | 15,78 | 319,17 |
| Four clusters: 3rd center (normal) | 3,34 | 316,90 |
| Four clusters: 4th center (normal) | 6,85 | 116,89 |
| Two clusters: 1st center (dip) | 3,02 | 80,37 |
| Two clusters: 2nd center (dip) | 5,48 | 331,56 |
| Three clusters: 1st center (dip) | 2,25 | 42,35 |
| Three clusters: 2nd center (dip) | 3,28 | 113,78 |
| Three clusters: 3rd center (dip) | 5,99 | 307,62 |
| Four clusters: 1st center (dip) | 2,15 | 58,06 |
| Four clusters: 2nd center (dip) | 4,02 | 355,53 |
| Four clusters: 3rd center (dip) | 3,39 | 122,03 |
| Four clusters: 4th center (dip) | 5,17 | 276,75 |



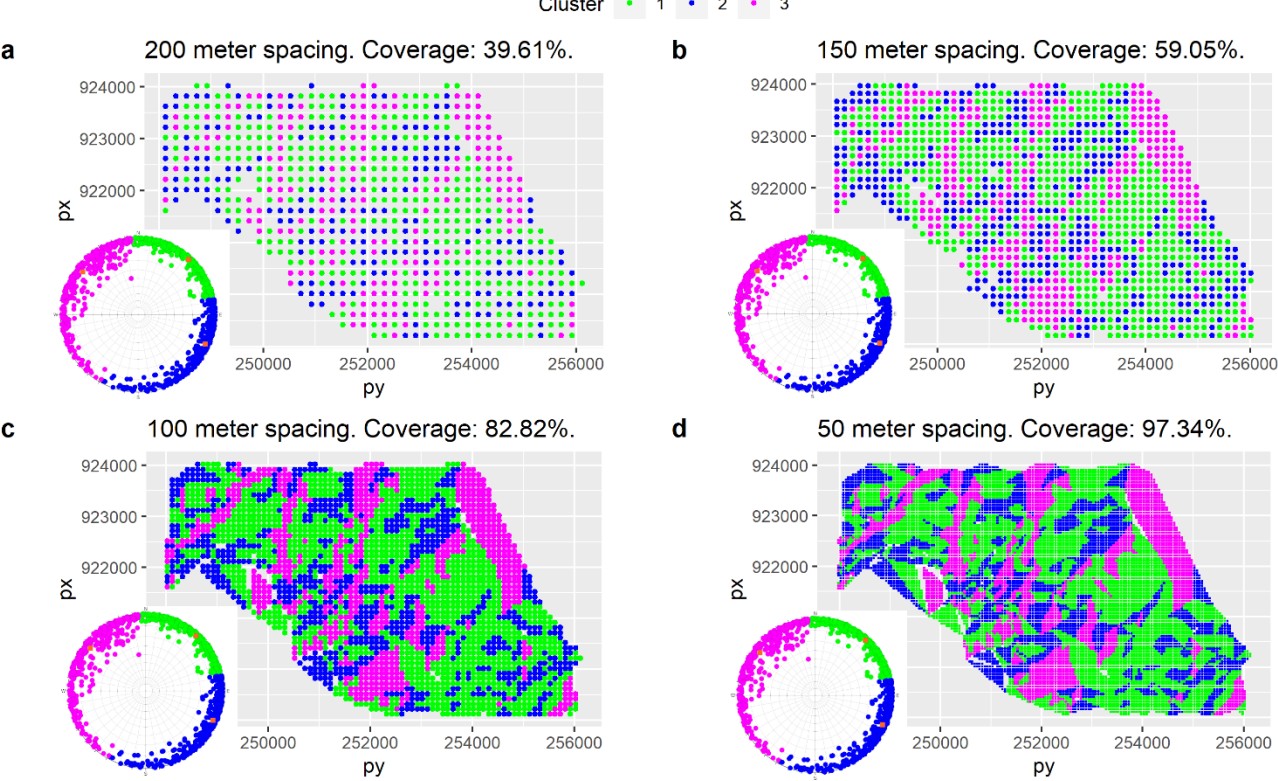

**Figure 14.** Regular version of spatial clustering. Impact of the grid density on the geometric granularity of the studied interface: (A) grid with points separated by 200 meters, (B) grid with points separated by 150 meters, (C) grid with points separated by 100 meters, and (D) grid with points separated by 50 meters. The coverage rates refer to the proportion of filtered (not including collinear) Delaunay triangles linked with the points from the regular grid to all filtered Delaunay triangles. Note that exceptions from the convex shape of the polygon or blank spaces in the interior are due to removed collinear configurations. We recommend to minimize the value of spacing to exhibit potential connectivity patterns.

# 7 Discussion

## 7.1 Method's capabilities

The method's promise to identify geometric anomalies lies in the fact that squared Euclidean distance inherent to the k-means algorithm (Hastie et al., 2009) is equivalent to cosine distance if processed vectors have unit length (Eq. 1-6). Thus, the resulting clusters have geometric meaning because they represent groups of observations that have small within-cosine distance, a fact often used in the field of text analysis (Choi et al., 2014; Zhang et al., 2020; Hornik et al., 2012). In structural studies, cosine distance as dissimilarity metric was used for detecting fracture sets in outcrops (Zhan et al., 2017a) based on



observations with substantial dip. In this study, we analyzed subsurface geological terrains with sub-horizontal or moderate dip and we believe that the method holds promise for providing insights into the subsurface architecture of similar class-imbalanced data, thus preventing the creation of oversimplified models (Caumon et al., 2009).

This study adds knowledge regarding the possibility of using computational geometry theorems to explain the meaning of resulting clusters. As a case in point, we note that clustering results have a repeatable pattern in that the normal representation produced almost collinear cluster centers (Fig. 15A) and the dip representation has always (except k=2 when there is no vertex and the theorem is not applicable) a vertex common for all clusters near the origin of the stereonet (Fig. 15B). These results can be rewritten using the computational geometry theorems. From Theorem 1 it follows that collinear cluster centers imply

parallel boundaries between clusters. Indeed, in our results, the approximate boundaries between clusters have a similar trend (Fig. 15A). This suggests that in our results, the boundaries between clusters may be distributed along a potential fold axis with cluster centers lying in the same plane which is perpendicular to the direction of a potential fold axis. Moreover, from Theorem 2 it follows, that a point q is a vertex in Voronoi diagrams if and only if its largest empty circle CP(q) contains three or more sites on its boundary. Thus, in Fig. 15B, the cluster centers must have a very similar distance to the origin which

implies that the cluster centers lie on a common circle and thus have a very similar dip angle (compare dip angles for dip representations in Tabs 1,2). A full explanation of the above effects lies beyond the scope of this research and needs further studies.





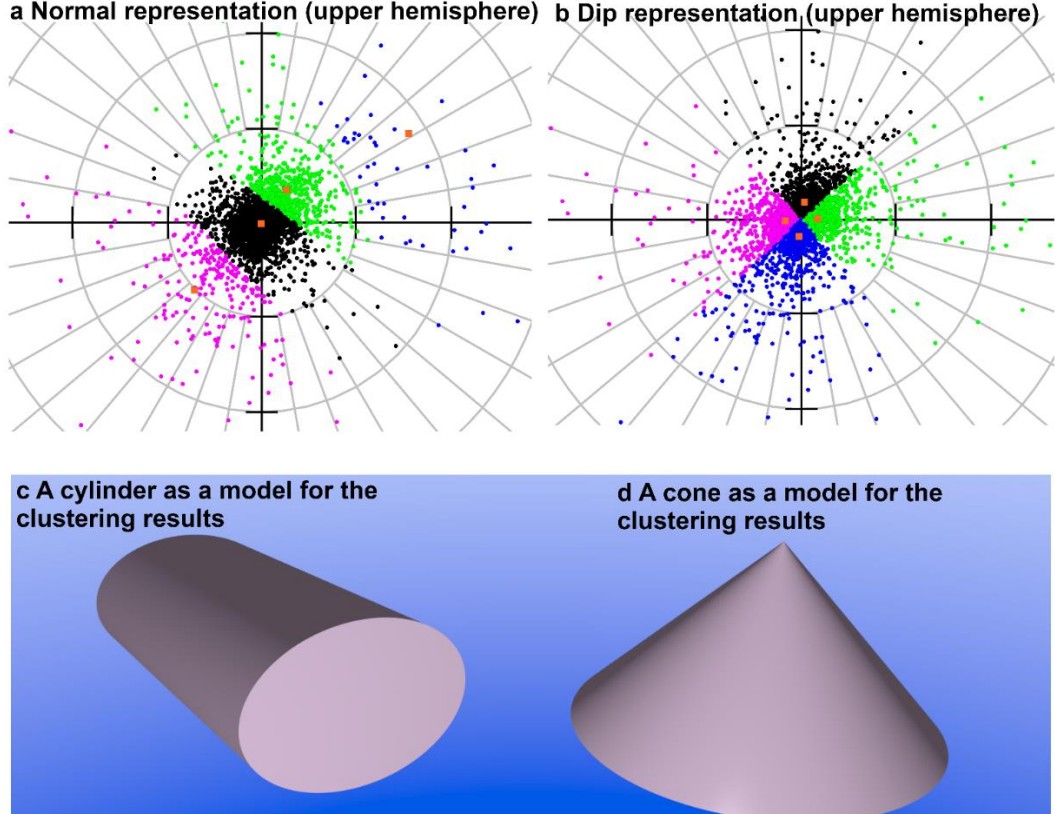

**Figure 15.** An observed clustering pattern: (A) when normal vectors are clustered, the cluster centers are almost collinear; Theorem 1 says that collinear cluster centers imply parallel boundaries between clusters; this result suggest that the boundaries between clusters may point to an axis of a potential megafold (B) when dip vectors are clustered, there is a common vertex for all clusters near the origin. This effect suggests (Theorem 2) that the centers are co-circular, which implies a common value of dip angle (compare dip angles for dip representations in Tabs 1, 2), thus pointing to a potential megacone (C) a cylinder as a model for the partitioning results of normal vector representation; (D) a cone as a model for the partitioning results of dip vector representation.

## 7.2 Regularization

The first version is irregular, which means that we assign an orientation label to the geometric center of a triangle. This arbitrary decision makes the resulting map biased. The second version reduces the arbitrariness by creating a regular point set that is linked with corresponding triangles and their orientation labels. This regularization may serve to solve topological problems, e.g., whether or not the hypothesized faults intersect. However, there are two caveats to the regularization step:

1) only the irregular version can provide additional insights into the observed features (e.g., explanatory hypotheses about drilling the fault plane or having a listric fault)





2)   The regular version is still sensitive to the initial spatial heterogeneity of the data density (the spatial configuration of the boreholes).

If the analysis of connectivity between anomalies is of interest, our recommendation is to minimize the spacing of the points in the grid so as to maximize the proportion of linked triangles. This should be helpful to better analyse connectivity between observed anomalies. However, a sparse grid could be potentially more useful in the generalization schemes related to upscaling frameworks (Carmichael and Ailleres, 2016).

## 7.3 Limitations

Assuming the spatial homogeneity of the subhorizontal dip to the NE (ideally a flat plane dipping to the NE), the distances between three points taken to construct the plane do not influence its orientation (it is the same plane). However, if this homogenous surface is faulted, then triangles genetically related to the faults have orientations different than that of the underlying faulted surface (Michalak et al., 2021). The directional within-dissimilarity of the triangles genetically related to faults (thus anomalies) may sometimes be high, with unintuitive dip directions of triangles opposite the fault dip direction

(Michalak et al., 2021). In addition, the dip angle of these triangles is affected by the density of the borehole network in the vicinity of the fault, with boreholes located closer to the fault resulting in relatively greater dip angles. This interplay between the initial technical conditions (density of boreholes) and tectonics may limit the epistemological value of the resulting interpretation based on dip-angle-informed clustering (e.g. Fig. 13A).

## 7.4 Expert-guided partitions

If the faults strike perpendicular to the preferred direction, then setting a threshold dip angle may be helpful to reduce the interpretational impact of the initial technical conditions (the density of the borehole network). For example, if faults strike perpendicular to the preferred fault direction (NE) and have their hanging wall to the NE, then the related triangles have dip angle values greater than that of the regional trend, irrespective of the spatial heterogeneity of boreholes. Therefore, we suggest that sometimes the assistance of expert knowledge is needed to answer more specific questions. Another argument in support

of the expert-guided partition is that although the combination of dip vector representation and the squared Euclidean distance metric can help identify the directional domains, in some cases, the underlying assumption may not be realistic because genetically related observations may dip to opposite directions. For example, if the fault strikes perpendicular to the preferred dip direction with the hanging wall lying to the SW, then triangles genetically related to these faults may dip to the SW when the points from the hanging wall and footwall are relatively close to each other (and/or with a high value of fault throw).

However, they may also dip to the NE at smaller angles, which may be the case if the distance between the points from the hanging wall and footwall is relatively high (and/or with a small value of fault throw), thus only flattening the general effect of the dip to the NE. Merging the two groups of observations with a dip direction difference of 180 is, however, unlikely if the combination of dip vector representation and the squared Euclidean distance (cosine distance for unit vectors) is applied (Figs.11B-13B).



We note that the space of the geometric hypotheses created by the cartographic results may be high and thus interpretationally challenging. We note that in the case of a lack of other geological knowledge or data, the method is capable only of indicating the strike of the hypothesized structures. This is because the dip direction associated with the triangles related to these faults may also be attributed to reverse faults that have dip directions opposite those of the related triangles (Michalak et al., 2021).

## 8 Conclusions

As Bond (Bond, 2015) argues, for much structural geology, it is fair to say that "it's all about geometry". The infinite three-dimensional space encountered in structural geology observations points to the need for generalization of geometric information to increase the capabilities of recognizing related structures and their topology (Kania and Szczęch, 2020; Thiele et al., 2016). We believe that the method holds promise for identifying the relationships between the effects of the forces that shaped the region and those that caused subareas to deviate from the regional plan (compare Davis, 2002). More detailed

conclusions are highlighted below:

- Compared to simply color-coding surfaces with respect to either dip angle or dip direction, our method allows investigation of the relationship between dip and dip direction anomalies. Moreover, in simple visualizations of dip angle and dip direction, the boundaries between colors in available default color palettes are established without considering an optimization criterion which may result in the lack of spatial integrity of the existing structures (Fig.

6C).

- In our approach, observations are separated according to an optimization criterion. Our method is capable of detecting geometric anomalies because applying squared Euclidean distance to unit vectors results in minimizing within-cluster cosine distance (Eq. 1-6). Obviously, the geometric meaning of the proposed optimization procedure can be completely lost if the processed vectors do not have unit length. Thus, we do not recommend scaling vectors according

to the size of the related triangles.

- In case of many sub-horizontal observations (which is true for many terrains), we propose two different conceptualization about the optimization procedure for normal and dip representations. For normal vectors representing sub-horizontal terrains, it is better to conceptualize the optimization as minimizing the within-cluster cosine distance. For dip vectors representing sub-horizontal terrains, it is better to conceptualize the optimization task

as maximizing the between-cluster cosine distance.

- The correspondence between Voronoi tessellation (Hastie et al., 2009) and clusters resulting from applying the k-means algorithm as well as computational geometry theorems allow to further explain the meaning of the clusters. Empirical results show that the combination of cosine distance with normal and dip vector representation holds promise for identifying axes of potential megafolds and slope of potential megacones, respectively (Fig. 15A-D).

These results should not however be extrapolated as a general rule for other study areas.

- The selection of triangulation as a source of collecting data for spatial clustering allows the internal structure of anomalies to be revealed. We created additional yet potentially competitive hypotheses about the nature of the observed anomalies, i.e., whether the internal structure of the singularity may be due to drilling a nonvertical fault plane or due to a wider deformation zone composed of many smaller faults.

## Code availability


Software for this research is available in these in-text data citation references (Michalak, 2021b).

**Name of code**: GeoAnomalia. **License**: GNU General Public License v3.0. **Developer**: Michał Michalak. **Contact address**: Institute of Earth Sciences, University of Silesia in Katowice, Poland. E-mail: michalmichalak@us.edu.pl. **Year first available**: 2021. **Hardware required**: Celeron CPU or better. **Software required**: Microsoft Visual Studio (2015, 2017,

2022). **Program language**: C++. **Program size**: 600 KB. **How to access the source code**: Available at: https://github.com/michalmichalak997/SurfaceCompare/blob/master/orientation_maps.cpp Setup guide: https://github.com/michalmichalak997/Triangulation_2/blob/master/README.md

## Data availability

Datasets for this research (input and processed data) are available in these intext data citation references: (Michalak, 2021a)

[available on request]

## Author contribution

MM devised the project, wrote the manuscript, performed the computations and discussed the results. LT discussed the results and created several hypotheses regarding the internal structure of the revealed anomalies. FW participated in the study conceptualization (formulating the idea of regularization) and discussed the results (creating hypotheses regarding the observed

anomalies). JŻ drafted the regional geological setting and discussed the results regarding the internal structure of the observed anomalies. KG participated in drafting the regional geological setting. MK advocated for the regularization and discussed the limitations regarding the interpretational value of dip-angle-based clustering. YPM described the setting and data from the Central European Basin System and discussed the results. PL participated in drafting the regional geological setting.

## Competing interests

The authors declare that they have no conflict of interest.



### Acknowldgements

This study was financially supported by the National Science Centre, Poland (2020/37/N/ST10/02504). The license of the Dips 8.0 software was purchased by the AGH University of Science and Technology, grant number 16.16.140.315. I thank Professor Janusz Morawiec for discussions about cones as models for clustering results.

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
