# Peer review of "Clustering has a meaning: optimization of angular similarity to detect 3D geometric anomalies in geological terrains"

_EGUsphere, 2022_

## Referee Comment (RC2)

**Clustering has a meaning: optimization of angular similarity to detect 3D geometric anomalies in geological terrains**

Michał P. Michalak1,2, Lesław Teper1, Florian Wellmann3, Jerzy Żaba1, Krzysztof Gaidzik1, Marcin Kostur4, Yuriy P. Maystrenko5, Paulina Leonowicz6

[referee-annotated manuscript omitted]

---

## Author Response (AR1)

We thank reviewers Dr Guillaume Duclaux and Professor Thomas Blenkinsop for their constructive criticism. Both reviews have a similar question: can the results (e.g. elbow method metrics, clustering on stereonets, structural interpretations etc.) be different for normal and dip vectors if these two representations convey the same geometric information? Our answer is: yes, and it can be demonstrated with Photograph 1 which shows differences in Euclidean distances and angles between dip and normal vectors of two subhorizontal observations (a black and a green notebook) that differ in their dip direction.

[Figure]

*Photograph 1: Two subhorizontal dip vectors (green and grey pens) with slightly different dip directions. Euclidean distance between these vectors is d1. Two subvertical normal vectors (both blue pens) of the same observations with the Euclidean distance between them d2. We can see that d1 is greater than d2 and a similar effect is applicable for angles a and b (the angle between dip vectors is greater than the angle between normal vectors: a>b).*

**Reviewer 1:**

This manuscript introduces a new workflow for dealing with geological-surface mapping using sparse subsurface data. In particular, this work develops and investigates two new features for geological mapping using unsupervised machine-learning : 1) the role of structural data representations (as normal and dips vectors) on clustering results, and 2) the characterisation of Voronoi diagrams to explain the meaning of the boundaries between obtained clusters. The potential of these two methods are illustrated through applications to a couple of examples focusing at the very large scale on clustering regular data for the bottom Jurassic surface of the Central European Basin System, and at a smaller scale on clustering of irregular data for a middle Jurassic interface within the Krakow-Silesian Homocline in South-Central Poland.

Now, I am definitely not an expert in either machine-learning, nor clustering methods... so I've reviewed this manuscript from the perspectives of a structural geologist to whom such methods could be very useful for interpreting subsurface geology and structures.

Overall, the manuscript is well written and organised, and seems well suited for EGUsphere readership. The application of the unsupervised clustering method is presented, tested and analysed for different k-means and different vector representations in numerous figures (that still require some editing and clarifications). Limitations are appropriately discussed which keeps the contribution very honest. Such new machine-learning approach will potentially provide opportunities for geologists to (re)interpret subsurface structures in regions with either available geological surfaces, or dense boreholes coverage. Based on my review - as a structural geologist - I would recommend accepting this manuscript after moderate revisions of the figures and minor revisions of the text.

I present below a few key points for which I have some questions/concerns followed by a list of minor comments.

**Comment #1**

1) Choice of the optimum number of clusters: I have some trouble understanding the k-means choices based on the elbow method the authors have employed to determine the optimal number of clusters in their case studies...

Clarify: The determination of the optimum number of clusters using the elbow method is about finding an inflection point in the sum of squares curve with $k$ (1,2,3… - number of clusters) on the horizontal axis and W(C) on the vertical axis, where W(C) is the sum of squared dystanse between observations being in the same cluster, C is a classification function that assigns labels to observations. For example, C(7) = 3 means that the 7th observation goes to the 3rd cluster.

Change in the manuscript:  we've improved caption to Fig. 7.

**References:**

Hastie, T., Tibshirani, R., and Friedman, J.: The Elements of Statistical Learning - Data Mining, Inference and Prediction, https://doi.org/10.1177/001112877201800405, 2009.

**Comment #2**

First, I wonder whether Figure 7 is flawed? Why y the y-axes values so variable between the normal and dip representations for a single dataset? Shouldn't the numbers be similar between a and b (CEBS), and c and d (KSH)?

Clarify: No, but this is a good question and a similar one was asked by the 2nd Reviewer Professor T. Blenkinsop (see his 3rd comment). Consider looking at Photograph 1 or the below spreadsheet (Fig. 1), which shows two subhorizontal observations v and u that were assigned to a common cluster (denoted by label 1) in both normal and dip vector representations. If you look at the below spreadsheet, you will see that the values of squared Euclidean distances (and the angles as well!) between subhorizontal observations v and u depend on the representation chosen for calculation. In the below example, the squared Euclidean distance between dip vectors is about 39 times greater than the squared Euclidean distance between normal vectors. It can be explained by Photograph 1 or an effect, that is illustrated in Fig. 2. Consider five pairs of unit observations with a constant directional separation 10 degrees (to honour the fact that dip and normal vectors point to the same direction) and changing dip angles (from subhorizontal to steeply dipping). These pairs of observations will intersect the hemisphere in different parts, some near the edge, other closer to the top of the hemisphere. It can be seen that the observations closer to the edge have a greater Euclidean distance (so also squared Euclidean distance) than those that are closer to the top of the hemisphere. This is not a perfect example for explanation (because normal and dip vectors should be on different hemispheres), but it can serve to illustrate that one should expect changes in squared Euclidean distances when rotating (or lifting) a subhorizontal dip vector to get a subvertical normal vector. In other words, subvertical representations of subhorizontal observations (in our case normal vectors of a subhorizontal surface) will have smaller squared Euclidean distances than the subhorizontal representations of the same observations.

| | A | B | C | D | E | F | G | H | I | J | K |
|---|---|---|---|---|---|---|---|---|---|---|---|
| 1 | | X_N | Y_N | Z_N | X_D | Y_D | Z_D | Dip_ang | Dip_dir | kmeans2n | kmeans2d |
| 2 | triangle v | 0,0140094 | 0,0145931 | 0,999795 | 0,69239 | 0,721239 | -0,02023 | 1,15913 | 46,1691 | 1 | 1 |
| 3 | triangle u | 0,00502711 | 0,00443314 | 0,999978 | 0,75001 | 0,661393 | -0,0067 | 0,38403 | 41,4073 | 1 | 1 |
| 4 | | | | | | | | | | | |
| 5 | | | | | | | | | | | |
| 6 | | between v_n and u_n | between v_d and u_d | c7/b7 | | | | | | | |
| 7 | squared Euclidean distance | 0,00018394 | 0,00708435 | 38,514501 | | | | | | | |
| 8 | dot product = cosine | 0,999908125 | 0,996457496 | | | | | | | | |
| 9 | 1 - dot product = 1- cosine | 9,18754E-05 | 0,003542504 | | | | | | | | |
| 10 | 2*(1-cosine) | 0,000183751 | 0,007085009 | | | | | | | | |
| 11 | angle | 0,776677405 | 4,824153424 | | | | | | | | |
| 12 | | | | | | | | | | | |

*Fig 1 This spreadsheet presents two observations v and u that are in the same cluster for both representations (normal vectors and dip vectors) when using the k-means algorithm. It can be seen that the squared Euclidean distance between dip vectors is about 39 times greater than the squared Euclidean distance between normal vectors.*

[Figure]

*Fig 2 This illustration presents synthetic data: pairs of observations with constant directional separation (10 degrees of separation, see on the left). On the right, a 3D view is presented which shows that the Euclidean distance between tips of the vectors decreases when the pairs are approaching the upper vertex of a hemisphere. This can serve to illustrate that k-means clusters with subhorizontal representations of observations (in our case: dip vectors of a subhorizontal surface) can indicate greater within dissimilarity than subvertical representations of observations (in our case: normal vectors of a subhorizontal surface). D1, d2, d3, d4 and d5 are Euclidean distances between vectors (distances between tips of these vectors) – if you square these numbers, you get squared Euclidean distances.*

Change in the manuscript: we improved caption to Fig. 7.

**Comment #3**

Each structural data provide both dip and normal values, for CEBS there should be 236380 data points. I might miss something related to the y-axis for each figure: what is "tot_withinss"?

Clarify: "tot_withinss" is the same as W(C) from the first comment: it is the sum of squared distances between observations in the same cluster.

Change in the manuscript: we improved caption to Fig. 7.

**Comment #4**

Why does Figure 7B can either have 2 or 4 optimum number of clusters (Line 285-286)?

Clarify: The elbow method sometimes suggests ambiguous results regarding the optimum number of clusters in that the rates of inflection in the curve are similar, so competitive options may appear equally tempting.

Change in the manuscript: according to the request of the 2nd Reviewer, we deleted maps presenting non-optimum numbers of clusters

**Comment #5**

The number of clusters will be very important for determining the data clustering pattern based on the cluster centers analysis, so I reckon this section should be strengthened.

Clarify. Please note that there are many competitive heuristics for determining the optimum number of clusters, so we are afraid that strengthening this section could result in a „false feeling of security".

Change in the manuscript: we've improved the caption to Fig. 7 (more details about the method).

**Comment #6**

2) Stereographic representations: This applies for figure 2c, 8, 9, 10, 11, 12, 13 and 14.

- On Figure 2c it isn't clear which hemisphere is displayed for the data + we don't know where the North is located on figures 2a and 2b.

Clarify: In Fig. 2C we used the lower hemisphere for dip vectors. North is parallel to Y axis.

Change in the manuscript: we added clarification regarding the hemisphere and representation in the caption of Fig. 2. We added clarification rearding North in Figs. 2a and 2b.

**Comment #7**

- Fig 8 to 14: There a lower and an upper hemisphere half globes shown next to the stereonets in all these figures. For the lower hemisphere the whole Stereonet is shown, not for the upper hemisphere...

Clarify: Indeed, we made a zoom on the upper hemisphere, because otherwise we would struggle to see the shapes of boundaries between clusters.

Change in the manuscript: none

**Comment #8**

what is the grid spacing on these stereo? I may have missed it but it isn't clear to me what projection is being used (I suppose an azimuthal polar stereographic projection, is that correct?) - structural geologist would classically use equal-area or equal-angle stereonets.  Please clarify this in the caption of Figure 8 where the stereos first appear.

Clarify: The grid spacing is 10 degrees for both, dip angles and dip directions. We applied a polar equal-angle stereographic projection.

Change in the manuscript: A clarification was added to captions.

**Comment #9a,b,c,d,e**

a)  Direct screenshots from Paraview are hard to read. I'm thinking of Figures 1, 2 and mostly 4 and 6. The scale are not always meaningful, or hard to read. For example in Figure 1 the color scale legend is scalars... I guess elevation would be more adequate.

Agree.

Change in the manuscript: In Fig. 1 we replaced „scalars" with „elevation", sizes of values of coordinates are larger now.

b) The bounding boxes scale units in figure 2 are not readable either

Agree.

Change in the manuscript: The scale units have been enlarged.

c) and are totally missing in Figure 4 and 6. I believe redrafting Figure 4 would massively improve its readability.

Agree.

Change in the manuscript: We've added units to Figs 4 and 6. We have redrafted Fig. 4. We've also removed dark background.

d) The moiré pattern visible in Figure b and c is just terrible.

Agree.

Change in the manuscript: The moiré pattern was removed from Figs. 4b,c.

e) Units should also be added to the scale bars, and finally the scale bars' min and max values should be written in the same encoding than the rest of the values (-7.2e+03 and 1.6e+03 should be -7200 and 1600; 1.6e-04 should be 0; 6.1e+01 should be 61; 2.2e-03 and 3.6e+02 should be 0 and 360).  Same goes for Figure 6.

Clarify: It is fine for us to add units, however, I cannot find an option in ParaView to adjust encoding to min max values and replace the min max values with the suggested ones – I would probably need to add an artificial data but I wouldn't like to do this.

Change in the manuscript:  we've added units but we couldn't adjust encoding. We didn't replace min max values with the suggested ones.

Minor comments:

**Comment #10**

+ Equation 2 page 4: why is each line for Eq. (2) development given a different number? This is the same equation and as such should be only referred to Eq (2).

Agree. We wanted to have a different number for each line in case of specific questions about the steps but we decided to follow your suggestion.

Change in the manuscript: we have now only Eq. 2.

**Comment #11**

+ Line 144: please remove the second "a" in this line: "[...] whether a specific 2D point p [...]"
Agree.

Change in the manuscript: the second „a" was deleted.

**Comment #12**

+ Line 234: "en échelon" is missing its accent.

Agree.

Change in the manuscript: Accent was added.

**Comment #13**

+ Line 240: Genus and species for *Strenocera subfurcatum* should be written in italic font.

Agree.

Change in the manuscript: italic added.

**Comment #14**

+ Line 272: Please revise the reference for the Anon borehole database citation. I understand it is not published, though.

Agree.

Change in the manuscript: We replaced „Anon" with „Unpublished"

**Comment #15**

+ Line 350-351: Theorem 1 --> Do you mean Eq 1?

Clarify: No, I meant Theorem 1 in section 3.3, which is about boundaries in a Voronoi diagram.

Change in the manuscript: none

**Comment #16**

+ Line 373-374: How would you differentiate between a graben structure and an "antithetic shear with hanging walls dipping against the main fault"?

Clarify: please note that since we have a „Method article", we only mentioned possible hypotheses for the observed effects. We believe that using this method we are not ready to attach likelihoods to these hypotheses.

Change in the manuscript: none

**Comment #17**

+ Line 440: missing s in "this result suggests"

Agree.

Change in the manuscript:  „s" added.

**Comment #18**

+ Figures and captions in general: the figures use lower case letter (a, b, c...) while in the captions and the text upper case letters are used (A, B, C...). Please harmonise between the figures and the text/captions.

Agree.

Change in the manuscript: we replaced (A, B, C...) with (a, b, c...)

Nice, 24/08/2022

Guillaume Duclaux

**Reviewer 2**

**Comment #1**

This is a difficult paper to read, because it contains a lot of jargon about geometry, and because of vague general statements, some of which are unnecessary (e.g. "dip angle is not capable of showing the dip direction of faults and vice-versa" and "Geology is considered to be a subjective science (Curtis, 2012)").

Agree. We wanted to emphasize that we do a three-dimensional analysis of outliers.

Perhaps it is better to give only examples of subjectivity without writing these general statements.

Change in the manuscript: we deleted the sentences:

- „Geology is considered to be a subjective science (Curtis, 2012)", and
- „dip angle is not capable of showing the dip direction of faults and vice-versa"

**Comment #2**

A further problem for understanding the paper is that some of the methods section is couched in the technical language of the CGAL library. This is unhelpful to the general reader, and needs to be explained in simple terms.

Clarify. This remark is about our chapter „3.4 Irregular and regular trend maps". We believe that not everyone needs these details, but we wanted to include as many details as possible to allow reproducibility of the regular version. Please note that the general message of this chapter is also presented in Fig. 2. We can summarise the method as follows:

„A summary of the regularization method: information about clustering labels of triangles must be attached to points from the regular grid. This transfer of information is possible via CGAL query functions which allow to identify triangles that have points in their interiors (the points are arguments of the query functions). Please note that executing the query functions and clustering are done in separate environments, therefore two datasets (Table X and Y) need to be merged using unique elements (ids of the vertices of triangles)."

Change in the manuscript: we improved Fig. 2 and caption to it

**Comment #3**

One of the main conclusions, that applying clustering methods to normal vectors and dip direction vectors from the same data set results in different interpretations of the structure (Fig. 15), seems unlikely to be correct. There is no material difference between the geometrical significance and information contained in a normal vector compared to a dip direction vector. If there is a difference in the outcome of the clustering methods, that must be an artefact of the way the methods have been applied to each data set.

Disagree/Clarify.

Disagree. We disagree that clustering results must be the same for the dip and normal vectors (see Photograph 1 or the explanation below). Please be informed that data sets with required data (coordinates of normal and dip vectors) are available to reviewers so that they can independently verify the results.

Clarify. Because of the differences in values of coordinates, we cannot assume that the squared Euclidean distance (which is the squared distance between tips of the vectors and which determines clustering results) calculated for two different representations of two observations will be equal.

Clarify. Consider Photograph 1 and the following examples:

I.      Intuitevely: If you have a subhorizontal surface, then the dip direction vectors will be subhorizontal as well, while the normal vectors will be subvertical (Fig. 1 in this file). And if subhorizontal dip direction vectors dip in opposite directions, then the distance between the tips of such vectors (d2) will be high compared to the distance between tips of subvertical normal vectors (d1) (see Fig. 2).

[Figure]

*Fig 1. An example showing that Euclidean distances (and then squared Euclidean distances) d1 and d2 for subvertical ( brown normal vectors) and subhorizontal (green dip vectors) representations of two observations can be different.*

II.     A spreadsheet with a numerical example similar to this in Fig. 1 (of this response file)

| | A | B | C | D | E | F | G | H | I |
|---|---|---|---|---|---|---|---|---|---|
| 1 | | X_N | Y_N | Z_N | X_D | Y_D | Z_D | Dip_ang | Dip_dir |
| 2 | triangle v | 0,0140094 | 0,0145931 | 0,999795 | 0,692391 | 0,721239 | -0,02023 | 1,15913 | 46,16906 |
| 3 | triangle u | -0,0217465 | -0,025059 | 0,999449 | -0,65506 | -0,754845 | -0,03318 | 1,901383 | 229,0482 |
| 4 | | | | | | | | | |
| 5 | | | | | | | | | |
| 6 | | between v_n and u_n | between v_d and u_d | | | | | | |
| 7 | squared Euclidean distance | 0,002850893 | 3,99462396 | | | | | | |
| 8 | dot product = cosine | 0,998573769 | -0,997312185 | | | | | | |
| 9 | 1 - dot product = 1- cosine) | 0,001426231 | 1,997312185 | | | | | | |
| 10 | 2*(1-cosine) | 0,002852462 | 3,994624369 | | | | | | |
| 11 | angle | 3,060442257 | 175,7982068 | | | | | | |

*Fig 2 An example showing differences in squared Euclidean distance calculated for unit normal and dip vectors of two observations v and u. It can be seen that the squared Euclidean distance between normal vectors of v and u is **0.002850893**, while the same distance between dip vectors of v and u is **3.99462396**.*

Change in the manuscript: none

**Comment #4**

Another main conclusion is that optimisation methods must be applied to investigate clustering. This is relatively trivial: any clustering algorithm requires a similarity index, and the one used here (cosine distance) is a standard metric for assessing orientation differences.

Clarify. We agree that clustering algorithms use similarity functions, but we didn't argue in the Conclusion that „optimisation must be applied to investigate clustering". Clustering in this case is optimisation, so the latter doesn't have to be „applied" to the former. Again, we didn't have such a sentence in the manuscript. We argued, however, that:

1) in the first bullet point of the Conclusion: that if you use a color pallette for dip angle or dip diretion available in the GIS software, then the boundaries between colors may be subjective and without optimization significance, so it may be better to use clustering (thus optimisation)
2) We argued that theorems about Voronoi diagrams are useful to explain meaning of the clustering results.

Change in the manuscript: none

**Comment #5**

Further to the previous point, this metric should not result in significant differences between normal and dip direction vectors, because the cosine distance between two normal vectors must be the same as the cosine difference between the two dip vectors of the same surface.

Disagree. We disagree that squared Euclidean distances, angles and cosine distances between two normal vectors (v_n, u_n) and two dip vectors (v_d, u_d) of two observations u and v must be the same – see Photograph 1. We are also a bit confused with the term „of the same surface" because if there are two normal and dip vectors, then in both cases they represent two distinct observations, so we would argue that they don't represent the same entity.

Clarify. In Fig. 3 (in this response file) you can see a couple of pairs of vectors with a constant directional separation (to honour that dip and normal vectors point to the same direction). The distances between tips of vectors that intersect the hemisphere are greatest (d1) at the bottom and lowest (d5) at the top of the hemisphere (the same applies to angles). This illustrates that if you rotate (or lift) a subhorizontal dip vector to get a subvertical normal vector, then you should expect changes in squared Euclidean distances, angles and cosine distances as well (see also Photograph 1).

[Figure]

*Fig 3 This illustration presents synthetic data: pairs of observations with constant directional separation (10 degrees of separation, see on the left). On the right, a 3D view is presented which shows that the Euclidean distance between tips of the vectors decreases when the pairs are approaching the upper vertex of a hemisphere. This can serve to illustrate that k-means clusters with subhorizontal representations of observations (in our case: dip vectors of a subhorizontal surface) can indicate greater within dissimilarity than subvertical representations of observations (in our case: normal vectors of a subhorizontal surface).*

Change in the manuscript: none

**Comment #6**

There is some discussion about anomalous results:

"The above effect could be explained by several competitive hypotheses. For example, the fault plane could have been drilled, 365 thus broadening the zone of triangles genetically related to the fault (Michalak et al., 2021). Assuming the tectonic origin of the related structures, it can be hypothesized that fault drags on the hanging wall contribute to subsidiary elevation differences that must be consumed by nearby triangles. It could also be argued that an unusual lowering of the contact surface is due to a deformation zone composed of many smaller faults. Another hypothesis could be that the related feature is not a fault but rather a sedimentary slope, which would explain the gradual lowering of the contact surface."

Such hypotheses are useful, but would be better illustrated with specific examples and some reasoning about which is the preferred hypothesis.

Clarify. We can agree that from the viewpoint of a structural geologist, it is better to select only one hypothesis and provide arguments. But please note that our paper is

classified as a „Method article", so we put „equal probability" to all possible hypotheses. Otherwise, we are afraid that the reviewers or readers could think that we aim to solve a specific geological problem, and it would no longer be a „Method article".

Change in the manuscript: none

**Comment #7**

The determination of the optimum number of clusters is explained in Figure 7, but the results sections shows results from 2, 3 and 4 numbers of clusters. This is unnecessary: only the optimum results should be shown.

Agree/Clarify. Figures related to non-optimum results such as 2 clusters for CEBS and 4 clusters for KSH can be indeed removed in both versions. In our opinion, some non-optimum results should remain (3 clusters for CEBS – dip vectors, 4 clusters for CEBS – normal vectors, 2 clusters for KSH – dip vectors, 3 clusters for KSH – normal vectors) because they are still useful for comparing representation results and for proposing models for clustering results.

Change in the manuscript: We decided to remove figures related to non-optimum results such as 2 clusters for CEBS and 4 clusters for KSH. Some non-optimum results remained (3 clusters for CEBS – dip vectors, 4 clusters for CEBS – normal vectors, KHS) because they are still useful for comparing representation results and for proposing models for clustering results.

**Comment #8**

The figures could be substantially improved. The use of such a dark background does not help (e.g. Fig. 6c). In most cases the grid is the most dominant and least important aspect of the maps, obscuring the detail of the clustering. The stereoplots are not explained in the figure captions.

Agree.

Change in the manuscript: We improved Figs 4, 6 and 8.

Other comments (in the annotated pdf):

We were requested by the Editor to address geological issues. It is possible that we will follow also other pieces of advice after discussing with Editors.

**Comment #9**

There is no need for these section headings in the Introduction

Clarify: we will discuss the issue with Editors because we believe that it can be helpful

Change in the manuscript: none

**Comment #10**

This is a trivial point whihc all strcutural geologist will understand, and none wouold make this mistake

Agree. We wanted to emphasise that we do a 3D analysis.

Change in the manuscript: the second part of the sentence was deleted

**Comment #11**

Figure reference out of order

Agree. But we would like to ask for an exception.

Change in the manuscript: none

**Comment #12**

Explain the colours and the colour bar. What units are these?

Clarify. Units correspond to elevation.

Change in the manuscript: figure was corrected

**Comment #13**

What is the CGAL library?

Clarify. Computational Geometry Algorithms Library – it contains many algorithms related to computational geometry.

Change in the manuscript: CGAL was expanded

**Comment #14**

This is the first time that boreholes have been mentioned. What boreholes are these?

Clarify. Well, the word „boreholes" is unnecessary because we also have geophysical surface data.

Change in the manuscript: „boreholes" deleted

**Comment #15**

Sentence does not make sense: replace as well as by and?

Agree.

Change in the manuscript: we replaced „as well as" by „and"

**Comment #16**

underwent

Agree.

Change in the manuscript: we replaced „were undergone" by „underwent"

**Comment #17**

In b and c the grid is much too prominent and obscures the data

Agree.

Change in the manuscript: We improved the figure.

**Comment #18**

Ores of what minerals?

Clarify. iron

Change in the manuscript: we added clarification

**Comment #19**

The clustering map is poor. Only one cluster memebrship ( te magneta one) can be clearly seen. The grid obscures much of this diagram

Clarify. We deleted this figure because it doesn't present optimum results – see your comment #7.

Change in the manuscript: we removed the figure

**Comment #20**

Wxplain the stereoplots. Why are there two versions in each figure? What advantage does the secind one have

Clarify. The stereonet on the left presents the projection of points from the unit lower hemisphere (tips of unit dip vectors) onto the horizontal plane. The stereonet on the right presents the projection of points from the upper hemisphere (tips of unit normal vectors) onto the horizontal plane. One advantage of using two versions is that in case of a subhorizontal surfaces, it is difficult to see boundaries of clusters when the combination of normal vector representation and upper hemipshere is used.

Change in the manuscript: we added explanations about stereonets in the captions

**Comment #21**

The grid is too prominent

Agree.

Change in the manuscript: we corrected the figure

**Comment #22**

I have no idea what this sentence means

Clarify. In case of triangles genetically related to discontinuities, dip angles are affected by the density of borehole network

Change in the manuscript: we added a clarification

**Comment #23**

This paragraph is extremely difficult to follow and would benefit from an additional diagram

Agree/Clarify. We agree that some rearrangements of the text are needed but we didn't include a new figure

Change in the manuscript: we've made some rearrangements of the text

**Comment #24**

Which method?

Clarify. We meant the general workflow.

Change in the manuscript: we replaced „methods" by „workflow"

**Comment #25**

This is not really a conclusion but just speculation.

Clarify. Because we don't speculate which version is „correct", we would argue that it is a conclusion

Change in the manuscript: none

We've introduced also our own changes:

**Own change #1 (the end of section 4.1):**

We added clarification regarding the details of the CEBS surface: "The investigated Jurassic horizon represents the base of the Jurassic in places where the Jurassic sediments are present (Maystrenko et al., 2013, 2012). Within the rest of the model area, this horizon corresponds to the top of pre-Jurassic sediments or to the top of the crystalline basement."

**Own change #2 (throughout the manuscript):**

We replaced „megafolds" with „megacylinders" because one can say about conical folds so using „megafolds" for cylinders only may be discriminative for conical folds

**Own change #3 (in discussion)**

We added a limitation regarding singular input data: „We note that dip vectors are not uniquely defined for horizontal observations (the dip direction cannot be specified), so we recommend removing horizontal observations prior to conducting clustering. A similar problem and proposed solution applies to normal vectors of vertical observations, for which two possible dip directions can be given."